# Immunogenic Properties of MVs Containing Structural Hantaviral Proteins: An Original Study

**DOI:** 10.3390/pharmaceutics14010093

**Published:** 2022-01-01

**Authors:** Layaly Shkair, Ekaterina Evgenevna Garanina, Ekaterina Vladimirovna Martynova, Alena Igorevna Kolesnikova, Svetlana Sergeevna Arkhipova, Angelina Andreevna Titova, Albert Anatolevich Rizvanov, Svetlana Francevna Khaiboullina

**Affiliations:** Open Lab “Gene and Cell Technologies”, Institute of Fundamental Medicine and Biology, Kazan Federal University, 420008 Kazan, Russia; L-Shkajr@stud.kpfu.ru (L.S.); EEGaranina@kpfu.ru (E.E.G.); AlIKolesnikova@stud.kpfu.ru (A.I.K.); SSArhipova@kpfu.ru (S.S.A.); AngATitova@kpfu.ru (A.A.T.); Albert.Rizvanov@kpfu.ru (A.A.R.); sv.khaiboullina@gmail.com (S.F.K.)

**Keywords:** orthohantavirus, hemorrhagic fever with renal syndrome, microvesicles, delivery system, vaccine

## Abstract

Hemorrhagic fever with renal syndrome (HFRS) is an emerging infectious disease that remains a global public health threat. The highest incidence rate is among zoonotic disease cases in Russia. Most cases of HFRS are reported in the Volga region of Russia, which commonly identifies the Puumala virus (PUUV) as a pathogen. HFRS management is especially challenging due to the lack of specific treatments and vaccines. This study aims to develop new approaches for HFRS prevention. Our goal is to test the efficacy of microvesicles (MVs) as PUUV nucleocapsid (N) and glycoproteins (Gn/Gc) delivery vehicles. Our findings show that MVs could deliver the PUUV N and Gn/Gc proteins in vitro. We have also demonstrated that MVs loaded with PUUV proteins could elicit a specific humoral and cellular immune response in vivo. These data suggest that an MV-based vaccine could control HFRS.

## 1. Introduction

Hemorrhagic fever with renal syndrome (HFRS) is a zoonotic disease characterized by an acute kidney insufficiency, hypotension, and coagulation abnormalities [1]. HFRS is caused by orthohantaviruses, which are transmitted by inhaling virus-contaminated aerosols [2]. It is a reportable disease, with the highest incidence rates in the Volga river region of Russia [3,4]. The Volga River region includes the Republic of Tatarstan, where hundreds of HFRS cases are documented annually [2]. In 2020, 3845 cases of hantavirus infection were registered in Russia, with the highest prevalence in the Volga region [5]. Moderate course of the disease was observed in 98.1% of cases, and 13 fatal outcomes were officially documented [5]. Although the mortality rate is low (0–0.4%), HFRS is frequently accompanied by serious complications involving the cardio-vascular and excretory systems [3,6].

The disease is likely to exhibit a cyclical pattern, with the highest and the lowest annual cases in humans observed every 3–5 years [4]. Between 2000 and 2017, a total of 131,590 HFRS cases were reported in 68 of Russia’s 85 administrative regions, an annual average rate of 4.9 cases/100,000 inhabitants. Annual incidence rates varied greatly, and epidemics occurred every 2–4 years with occasional 2-year peaks, such as 2008–2009 and 2014–2015 [2]. According to the state report of Rospotrebnadzor in Russia, the economic damage caused by viral hemorrhagic fevers amounted to more than USD 2 million [7]. It was reported that the incidence rate increased by 60% as compared to that in 2018. The high incidence rate and prevalence among the young calls for the development of preventative measures against this disease. Currently, there is no approved vaccine in the Russian Federation, therefore disease management remains focused on treating the symptoms [8,9,10].

Puumala virus (PUUV), a member of the *Orthohantavirus* genus, is commonly found as an HFRS pathogen in the Republic of Tatarstan [2]. Orthohantaviruses (order *Bunyaviriales*) have a negative-sense, single-stranded RNA genome [11]. The genome contains three segments—L, M, and S—which encode for RNA-dependent RNA polymerase (RdRp), envelope glycoproteins (Gn and Gc), and a nucleocapsid (N) protein, respectively [12,13]. It was shown that N and Gn/Gc could elicit specific immune responses [3,14,15,16,17], and they have been used as components of several vaccines [18,19,20,21,22,23]. The first hantavirus vaccine, Hantavax, was developed in South Korea in 1990 [24]. This vaccine contained inactivated viruses derived from the brains of rodents and cell cultures. Hantavax was reported to elicit antibody responses detectable by ELISA, but neutralizing antibodies were detected in only about half of the recipients [25]. By contrast, in Korea, only half of the vaccinated individuals in a cohort had anti-HTNV N and anti-HTNV Gc protein-neutralizing antibodies [26]. Later, four inactivated vaccines were developed in China; however, these vaccines had a low immune reactivity [27,28]. Commercially available inactivated Hantaan (HTNV) and Seoul orthohantavirus vaccines used in China were shown to induce a humoral immunity in 2013 [29]. Increased levels of anti-HTNV N protein specific IgM, IgG, and anti-HTNV G protein-neutralizing antibodies were detected in the sera of vaccinated individuals [29]. The immune response was effective in reducing incidences of orthohantavirus infections [29,30,31], but vaccination failed to reduce the disease severity [32]. Therefore, it appears that the vaccines induced only short-term immune responses that required booster immunizations. Currently in the Russian Federation there is no approved vaccine for HFRS prevention. In 2019, there were reports of successful pre-clinical studies of an inactivated polyvalent vaccine based on PUUV, HTNV, and Dobrava–Belgrade strains [28,33]; nevertheless, the vaccine is not available for general vaccination.

Another approach to developing protection against orthohantavirus infections is the use of epitope peptide vaccines. Using an immunoinformatic approach, Abdulla et al. proposed an ideal formula for orthohantavirus vaccines in which viral peptides carrying immunogenic epitopes were conjugated with TLR-4, a natural adjuvant [34]. The efficacy of this vaccine remains unknown. In 2020, Ma et al. designed a macromolecular HTNV multiple antigenic peptide (MAP) with a polylysine core and single-CTL-epitope chains [35]. It was demonstrated that HTNV MAP could stimulate the CD8+ T cell secretion of IFN-γ in HLA-A*02 + HFRS patients [35]. In another approach, the efficacy of DNA vaccine coding for the Gn/Gc of orthohantaviruses to induce the antibody response was demonstrated [36,37,38]; yet the immunogenicity of these vaccines was shown to be inconsistent [36]. Improving the efficacy of orthohantavirus vaccines remains central to developing safe and effective vaccines.

It has previously been demonstrated that HFRS patients gain long-lasting protection following infection [39,40]. Neutralizing antibodies have been detected years after PUUV, SNV, and ANDV infections, indicating a long-lasting immune response [41,42]. Notably, the reactivity of antibodies in HFRS convalescent sera were detected in both Puumala (PUUV) and Hantaan viruses [43,44]. Natural infection could be associated with an abnormal kidney function (a higher prevalence of tubular proteinuria and increased urinary protein excretion) and cardiovascular disturbances even 6 years after acute HFRS [45,46]. Therefore, it is very important to develop novel strategies providing cross-protective immunity.

Recently, membrane microvesicles (MVs) have gained attention as potential vaccine delivery tools [47,48]. MVs are nano-sized extracellular vesicles, physiologically produced by all cell types through direct shedding from the plasma membrane [49,50,51,52]. There are multiple advantages of MVs as a delivery system, including their endogenous origin, stability, and ability to cross natural biological barriers, which make them favorable for a wide range of applications [47]. One of these applications is their use as a delivery vehicle for natural and synthetic substances [53,54]. The delivery potential of MVs was verified by the demonstration of the specific immune response activation against their cargo through the uptake of the antigen by antigen-presenting cells [55,56].

Based on the evidence supporting the successful vaccine delivery using MVs [57,58,59,60], we sought to determine their efficacy in the following ways: 1. carrying and transferring orthohantavirus PUUV N and Gn/Gc proteins in vitro; and 2. eliciting humoral and T cell immune response in vivo. We have demonstrated that MVs could deliver PUUV N and Gn/Gc proteins in vitro. We also found that MVs carrying PUUV proteins could produce a specific humoral and cellular immune response in vivo. Additionally, the activation of CCL11, TNF-α, IFN-γ, IL-6, G-CSF, and GM-CSF in mice treated with MVs taking PUUV N and Gn/Gc proteins was demonstrated at 14 and 28 days following treatment.

## 2. Materials and Methods

### 2.1. Cells

Murine mesenchymal stem cells (mMSCs) were isolated from C57BL/6 mice adipose tissue [61]. Slices of adipose tissue were washed (0.9% NaCl, PanEco, Moscow, Russia) and digested using 0.4% collagenase (Biolot, Saint Petersburg, Russia) for 1 h at 37 °C with vigorous shaking at 220 rpm. mMSCs were pelleted (1500 rpm, for 5 min), washed (3×; PBS) and maintained using Dulbecco’s modified Eagle’s medium (DMEM, PanEco, Moscow, Russia) supplemented with 10% fetal bovine serum (FBS, HyClone, Utah, UT, USA) and 2 mM L-glutamine (PanEco, Moscow, Russia) at 37 °C with 5% CO_2._

Human embryonic kidney HEK293T cells and human lung cancer A549 cells were obtained from the American Type Culture Collection (ATCC) and maintained in DMEM (PanEco, Moscow, Russia) supplemented with 10% fetal bovine serum (FBS, HyClone, Utah, USA) and 2 mM L-glutamine (PanEco, Moscow, Russia) at 37 °C with 5% CO_2._

### 2.2. Animals

Female C57Bl/6 mice (8–10 weeks old) were obtained from the animal breeding facility of the Kurchatov Institute (Rappolovo, Russia). Mice were housed at the Kazan Federal University animal facility. All animals received food and water ad libitum and were contained in a room with a 12 h light/12 h dark cycle and an ambient temperature of 22 °C. Mice received subcutaneous injection of MVs (15 µg/50 µL). Selection of MV concentration was based on previous publications according to which the concentration 15 µg/mouse was sufficient for induction of immune response [62,63]. The Institutional Ethics Committees of the Kazan Federal University approved procedures including the use of animals (protocol #23. 06.30.2020).

### 2.3. PUUV Peptides

Peptides (95% purity) were synthesized by GeneScript (Jiangsu, China) using the N and Gn/Gc protein amino acid (aa) sequence (Genbank accession numbers: CAA43370.1 and AAB01661.1, respectively). The sequence of peptides is summarized in Table 1.

### 2.4. Flow Cytometry

mMSCs (5  ×  10^5^ cells) were incubated with Alexa Fluor 647 anti-mouse CD73 antibodies (BioLegend, San Diego, CA, USA), Brilliant Violet 421 anti-mouse CD90.2 antibodies (BioLegend, San Diego, CA, USA), PE anti-mouse CD49 antibodies, and PE anti-mouse-CD29 antibodies (BioLegend, San Diego, CA, USA). Washed (3×, PBS) cells were analyzed using flow cytometry with a FACS Aria III (Becton, Dickinson and Company, Becton Drive Franklin Lakes, Franklin lakes, NJ, USA). A minimum of 300,000 events were collected for each sample. The fraction of mMSCs was selected using FSC-A (forward side scatter) and SSA (side scatter) parameters based on cell size, granularity, and viability. To reduce contamination with cell debris and exclude auto-fluorescence, we selected only cells with mMSC characteristics. Non-stained mMSC cells were used for gating as shown in Appendix A. mMSCs were identified as being Sca-1^+^, CD90^+^, CD73^+^, CD29^+^, and CD49^−^ phenotypes.

### 2.5. Recombinant Lentivirus

Recombinant lentiviruses (LV) were generated by transfecting HEK293T cells with a mix of packaging (psPAX2; 6.63 µg; Addgene, Watertown, MA, USA), envelope (pCMV-VSV-G; 3.51 µg; Addgene, Watertown, MA, USA) plasmids as well as one of the plasmids encoding for PUUV N (pLX-PUUV-S; 10.14 µg) or Gn/Gc (pWPT-PUUV-M; 10.14 µg) proteins. Plasmid vectors pLX-PUUV-S and pLX-Katushka2S were previously generated using Gateway cloning technology, while pWPT-PUUV-M was subcloned by GenScript (GenScript, Jiangsu, China).

The medium was replaced 16 h after transfection and supernatant was collected every 12 h for 96 h. Cell debris was removed (5000× *g*, 2 min) and supernatant was concentrated by ultracentrifugation (26,000 rpm, 4 °C, 2 h). Recombinant lentivirus aliquots (200 µL) were stored at −80 °C. The lentiviruses and proteins coded are summarized in Table 2.

### 2.6. Transduction

mMSCs were seeded (6 × 10^5^ cells/flask) and maintained in DMEM supplemented with 10% fetal bovine serum (FBS) and 2 mM L-glutamine at 37 °C with 5% CO_2_. mMSCs were transduced with recombinant lentivirus (LV-PUUV-S, LV-PUUV-M, and LV-Katushka2S) carrying PUUV N, Gn/Gc, or red fluorescent protein Katushka2S (MOI 100) using Opti-MEM medium (Gibco, Grand Island, New York, USA) supplemented with 8 µg/mL protamine sulphate (Sigma, Saint Louis, USA). Fresh media was added after 24 h to support cell proliferation. The medium and cells were collected 48 h later.

### 2.7. MVs Production

Genetically modified mMSCs were trypsinized, washed (2×, PBS; PanEco, Moscow, Russia) and incubated in serum-free DMEM/F12 (PanEco, Moscow, Russia) medium supplemented with 10 µg/mL Cytochalasin B (Sigma, Sant Louis, USA) for 30 min (37 °C, 5% CO_2_). Cells were vortexed for 60 sec to produce MVs. Cells were separated (700 rpm, 10 min) and MVs were collected by two subsequent centrifugations of the supernatant (700 rpm for 20 min and 15,000 rpm for 25 min).

### 2.8. Transmission Electron Microscopy (TEM)

MVs were fixed (2.5% glutaraldehyde; Sigma, Sant Louis, MI, USA) in 10% neutral buffered formalin (BioVitrum, Saint Petersburg, Russia) for 2 h and incubated in 1% osmium tetroxide (Sigma, Hamburg, Germany) for 1 h in the dark. Then, MVs were dehydrated in ethanol gradient (30–50–70%) followed by acetone (30 min) and oxypropylene (30 min). Dry MVs were impregnated with epoxy resin mixed with propylene in the proportion of (2:1–1:1–1:2), and each was incubated for 12 h or overnight. Then, MVs were incubated in 100% epoxy resin for 1 h 30 min at room temperature (RT) and then polymerized for three days at 37°, 45°, and 60°. Then, MVs samples were cut using a Leica EM ultramicrotome (Leica UC7, Wien, Austria) and mounted on a grid. MV sections were contrasted by uranyl acetate and lead citrate and were scanned using a TEM (Hitachi HT7700 Exalens, Tokyo, Japan). The diameters of MVs in each experimental group were calculated individually (5 images per each group) using ZEN 2.3 (blue edition)software (Carl Zeiss Microscopy GmbH, Jena, Germany).

### 2.9. Protein Concentration Analysis

Total protein concentration was determined using a BCA protein kit (Pierce, Rockford, IL, USA) according to the recommended instructions. Briefly, MVs (150 μL) were lysed in buffer (50 mM Hepes, 150 mM NaCl, 1 mM EDTA; 0.5% Triton X100, 10% glycerol; Panreac, Barcelona, Spain) containing Halts protease inhibitor (1 μM; Thermoscientific, Rockford, IL, USA) for 30 min and pelleted (12,000× *g*). Supernatant containing proteins was collected and stored at −80 °C.

### 2.10. Multiplex Analysis

Cytokine and chemokine analysis was done using the Bio-Plex Pro Mouse Cytokine 23-plex Assay (BioRad, Hercules, CA, USA) according to the manufacturer’s protocol. Twenty-three cytokines and chemokines (IL-1α, IL-1β, IL-2, IL-3, IL-4, IL-5, IL-6, IL-9, IL-10, IL-12 p40, IL -12 p70, IL-13, IL-17A, Eotaxin (CCL11), G-CSF, GM-CSF, IFN-γ, Gro-a (CXCL1), MCP-1 (CCL2), MIP-1α (CCL3), MIP-1β (CCL4), RANTES (CCL5), and TNF-α) were analyzed. Sample aliquots (50 μL) were used in the study with a minimum of 50 beads per analyte acquired. Median fluorescence intensities were collected using a MAGPIX analyzer (Luminex, Austin, TX, USA). Each sample was analyzed in triplicate. Data collected were analyzed with MasterPlex CT control software and MasterPlex QT analysis software (MiraiBio, San Bruno, CA, USA). Standard curves for each cytokine were generated using standards provided by the manufacturer.

### 2.11. Cell Treatment with MVs

A549 cells were seeded (5 × 10^4^ cells/well) for 24 h to generate a 70% monolayer. MVs (30 μg/well) containing PUUV proteins were applied onto the A549 cell monolayer for 48 h. Then, the A549 cell monolayer was washed (3×, PBS) before protein collection.

### 2.12. Western Blot

Proteins (10 µg) were loaded into wells of 12% gel, separated using electrophoresis and transferred onto polyvinylidene difluoride (PVDF) membranes (BioRad, Hercules, CA, USA). Membranes were blocked (5% non-fat dry milk, PBS + Tween 0.1% (PBS-T)) and incubated in rabbit anti-N protein (1:500, Sigma, Darmstadt, Germany) or mouse anti-Gc protein (1:500, Abcam, Waltham, MA, USA) antibodies. Secondary anti-rabbit HRP or goat anti-mouse IgG-HRP (1:2000; Sigma, Darmstadt, Germany) antibodies were used to form antibody–antibody complexes, visualized using Clarity ECL Substrate solution (BioRad, Hercules, CA, USA). Images were captured with ImageLab Software.

### 2.13. Immunofluorescence Assay (IFA)

The A549 cell monolayers were fixed (methanol, 10 min), rehydrated with PBS and permeabilized using 0.1% Triton X-100 solution for 20 min. Monolayers were incubated with primary rabbit anti-N protein (1:500; Sigma, Darmstadt, Germany) and mouse anti-Gc protein (1:50; Abcam, Waltham, MA, USA) antibodies (3 h, RT) followed by donkey anti-rabbit IgG (H + L) Alexa Fluor 647 (1:1000, Invitrogen, Rockford, USA) and donkey anti-mouse IgG (H + L) Alexa Fluor 555 (1:1000, Invitrogen, Rockford, USA), respectively. The nucleus was visualized using 4′,6-diamidino-2-phenylindol (DAPI; Sigma, Darmstadt, Germany). Images were captured using laser confocal microscope LSM 700 (Carl Zeiss) and ZEN software.

### 2.14. Enzyme-Linked Immunosorbent Assay (ELISA)

Analysis of serum IgG was done using a VektoHanta IgG ELISA kit (Vektor Best, Ufa, Russia) with a few modifications. A serum sample (50 µL; 1:100) was added to the wells pre-coated with orthohantavirus antigens (1 h at 37 °C), washed (5×, PBS-T) and incubated with goat anti-mouse IgG-HRP (1:10,000; American Qualex technologies, San Clemente, CA, USA) secondary antibodies (30 min at 37 °C). Immune complexes were visualized by adding 3, 3′, 5, 5′-tetramethylbenzidine solution (Khemamedica, Moscow, Russia). Results were obtained using a Tecan 200 plate reader (Tecan, Hombrechtikon, Switzerland) at OD_450_.

### 2.15. ELISpot Assay

ELISpot analysis was done using a mouse IFN-γ ELISpot kit (Abcam, Waltham, MA USA). Murine lymphocytes obtained from inguinal lymph nodes (1 × 10^5^ cells/well) were loaded into a well of 96 well plates. Lymphocytes were stimulated with each peptide (1 µg/well) for 24 h at 37 °C, 5% CO_2_. Peptide sequences are summarized in Table 1. Lymphocytes treated with phytohemagglutinin (1 μg, PHA; PanEco, Moscow, Russia) were used as positive control. Cells without treatment were used as the negative control. Each peptide was analyzed in duplicate.

### 2.16. Statistical Analysis

Statistical analysis was done using version 8.0.1 of GraphPad Prism software (244). The statistical significance was determined using one-way ANOVA with the multiple Tukey’s and Dunnett’s tests: ns = not significant; * *p* < 0.05; ** *p* < 0.01; *** *p* < 0.005; and **** *p* < 0.0001. The *p* value < 0.05 was considered statistically significant.

## 3. Results

### 3.1. Immunophenotype of mMSCs

mMSCs were isolated from adipose tissues of C57BL/6 male mice and analyzed using flow cytometry (Figure 1). mMSCs (95% of the whole cell population) were positive for the expression of markers intrinsic to MSCs: CD29 (β-integrin, 95.1%), Sca-1 (murine hematopoietic and mesenchymal stem/progenitor cell marker, 96.4%), and CD73 (95.4%), CD90 (Thy-1, 95.1%). Cells were negative for the CD49 marker (α5 integrin, 0.5%), also indicating their mMSCs origin [64]. The percentage of cells simultaneously expressing CD29 and Sca-1 was 90%, while 95.9% were CD73^+^CD90^+^.

### 3.2. TEM Analysis of MVs Size and Structure

mMSCs were transduced with lentiviruses expressing PUUV N (LV-PUUVS), Gn/Gc (LV-PUUV-M), a combination of N and Gn/Gc proteins (LV-PUUV-S and LV-PUUV-M), and a fluorescent protein (LV-Katushka2S). MVs were obtained 48 h after transduction by using the cytochalasin B treatment followed by a series of subsequent centrifugations of the supernatant [65]. MVs from non-transduced mMSCs served as the control. The size and structure of mMSC-derived MVs was captured using TEM (Figure 2). We found that the MVs had a round shape (Figure 2A) and diameters varying from 100 to 1000 nm (Figure 2B), which is the expected size of MVs [66]. We analyzed the size distribution of MVs carrying different PUUV proteins to demonstrate that MVs carrying different PUUV proteins maintained the size commonly identified with MV characteristics and that the PUUV protein cargo did not affect the MVs’ size.

### 3.3. Western Blot Analysis of MVs

We sought to determine the PUUV protein load in MVs derived from mMSCs transduced with lentiviruses expressing PUUV N (LV-PUUV-S), Gn/Gc (LV-PUUV-M), and a combination of N and Gn/Gc proteins (LV-PUUV-S and LV-PUUV-M) (Figure 3). PUUV N and Gn/Gc proteins were detected in a cargo of MVs.

### 3.4. Cytokines Released by Genetically Modified mMSCs

Twenty-three serum cytokines were analyzed by multiplex assay (BioRad, Hercules, USA). Initially, we determined the cytokine and chemokine levels in transduced mMSCs media (Figure 4 and Appendix A). Cells transduced with LV-Katushka2S were considered the reference control.

Interestingly, an increased level of CXCL-1 was demonstrated in supernatant of mMSCs expressing combinations of PUUV proteins only, indicating links to PUUV protein-expression. Simultaneous expression of PUUV N and Gn/Gc appears to have a more profound effect on cytokines activation as increased levels of CCL2 and G-CSF was found in supernatants of mMSCs.

### 3.5. Analysis of Cytokines in Cargo of Genetically Modified mMSCs

Next, we analyzed cytokines in MVs cargo (Figure 5 and Appendix A). Interestingly, we found more cytokines upregulated in cargo of MVs carrying PUUV proteins as compared to the supernatant of the parental cells. Additionally, CCL2, CXCL1 and G-CSF were upregulated in supernatant of mMSCs transduced with LV-PUUV S and M, levels of IL-2, IL-6, IL-13, IL-17A, CCL2, CCL3, CCL4, CCL5, CCL11, CXCL1, GM-CSF and TNF-α were increased in MVs cargo.

Many of these cytokines (IL-6, CCL2, CCL4, CCL5, CXCL1 and GM-CSF) were upregulated in MVs expressing PUUV proteins. This data suggests that MVs cargo contains a plethora of cytokines and chemokines which are present together with PUUV proteins. We suggest that these cytokines, delivered together with PUUV proteins, could attract leukocytes and stimulate leukocytes facilitating antigen uptake.

It is worth pointing out that upregulated levels of IL-2, CCL3, CCL4, G-CSF and TNF-α were detected in MVs carrying PUUV N or PUUV N and Gn/Gc, while increased concentrations of CCL11 and GM-CSF was observed only in MVs containing both hantaviral proteins. Thus, the combination of N and Gn/Gc proteins is likely to have more immunogenic potential.

### 3.6. Analysis of PUUV N and Gn/Gc Proteins Expression in A549 Cells Treated with MVs

To demonstrate PUUV protein transmission, A549 cells were treated with MVs derived from mMSC expressing PUUV N, Gn/Gc or a combination of N and Gn/Gc proteins. Cells were harvested 48 h after MVs treatment and analyzed using western blot. Specific bands representing PUUV N (50 kDa) and PUUV Gc (56 kDa) proteins [37] were demonstrated (Figure 6) in A549 cells treated with MVs suggesting the transfer of these PUUV proteins from MVs.

Expression of PUUV proteins in A549 cells treated with MVs from mMSCs was also demonstrated using IFA (Figure 7). PUUV N and Gn/Gc proteins were detected in A549 cells treated with MVs carrying PUUV N and Gn/Gc, suggesting the transfer of orthohantavirus proteins from MVs.

### 3.7. Serum Anti-PUUV Antibodies

Mice were injected subcutaneously with MVs (15 µg/50 µL) containing one of the following: PUUV N protein, PUUV Gn/Gc protein, or PUUV N and PUUV Gn/Gc proteins. Animals treated with physiological solution (50 µL; 0.9% NaCl) and mice treated with MVs-Katushka2S were used as control. The concentration of MVs (15 µg) showed a significant elevation in comparison to control groups. However, there was no significant difference between (10 µg, 15 µg, 20 µg, 30 µg) concentrations (Appendix A). There were eight mice in each group. To monitor immune responses to PUUV proteins, serum samples were collected at 14 and 28 days after MVs administration. Anti-orthohantavirus specific IgG were detected using VektoHanta IgG ELISA (Vektor Best, Russia). A significant elevation of anti-orthohantavirus IgG was found in mice treated with MVs carrying PUUV N and Gn/Gc at 14-days as compared to MVs-Katushka2S (Figure 8A). Increased levels of anti-orthohantavirus serum IgG in mice treated with MVs carrying PUUV N or Gn/Gc as well as combination of these proteins was found 28-day after treatment, as compared to MVs-Katushka2S (Figure 8B).

### 3.8. Cytotoxic T-Cell Activation

T cell activation is essential for the efficacy of anti-viral vaccines [67,68] providing specific defense and memory against the pathogen [69,70,71]. Therefore, we sought to determine whether MVs can activate T cell immune response. Several N and Gn/Gc proteins peptides were used to stimulate leukocytes from mice treated with MVs carrying PUUV proteins. Cytotoxic lymphocyte activation was analyzed by counting spots produced by INF-γ, released by lymphocytes, reacting with anti-INF-γ antibodies on membranes. Peptides used in this study are summarized in Table 1. PUUV N (N25, N29 and N43), Gn (M26 and M44), and Gc (M82) peptides stimulation increased INF-γ secretion by leukocytes from mice received MVs containing combined PUUV N and Gn/Gc proteins at 14 days compared to MVs-Katushka2S (Figure 9A). Only the M26 peptide was activating leukocytes from mice treated with MVs carrying PUUV Gn/Gc and MVs-PUUV N and Gn/Gc proteins at 14 days, compared to MVs-Katushka2S, while M44 and M82 peptides were activating leukocytes only in MVs-PUUV N and Gn/Gc. We also found the activation of cytotoxic T-lymphocytes in mice treated with MVs carrying PUUV N as well as a combination of PUUV N and Gn/Gc proteins after stimulation with N25 peptide at 28 days as compared to Katushka2S (Figure 9B). Interestingly, N25, N29, and N43 peptides provoked activation of cytotoxic leucocytes only in MVs-PUUV N and Gn/Gc after 14 days, while after 28 days significant IFN-γ secretion was detected after stimulation with N25 both in MVs-PUUV N and MVs-PUUV N and Gn/Gc groups. Activation of cytotoxic leukocytes after stimulation with N29 was pronounced only in the MVs-PUUV N group after 28 days. In contrast, peptides N43, M26, M44 and M82 had limited effect on secretion of IFN-γ by leukocytes collected from mice treated with MVs carrying PUUV N and Gn/Gc proteins on day 28 as compared to Katushka2S (data not shown).

### 3.9. Serum Cytokine Analysis

The immune response is regulated by cytokines that promote phagocytosis, attract leukocytes, and stimulate leukopoiesis [72,73,74,75]. Since MVs were carrying PUUV proteins together with cytokines, we proposed that this cargo could trigger activation and release of cytokines within the host. Therefore, we analyzed serum levels of twenty-three cytokines in mice that received MVs at 14 and 28 days after MVs treatment as compared to MVs-Katushka2S (Figure 10 and Appendix A). We found increased serum levels of TNF-α in animals that received MVs loaded with PUUV N protein at 14 days, as compared to animals treated with MVs-Katushka2S. Interestingly, MVs loaded with PUUV Gn/Gc significantly decreased levels of CCL3 in MVs-PUUV Gn/Gc groups; while CCL11 was upregulated (Figure 10A) at 14 days as compared to MVs-Katushka2S.

At 28 days, levels of IL-6, G-CSF and GM-CSF were increased in serum of animals treated with MVs loaded with PUUV N protein as compared to control (Figure 10B). Elevated levels of IFN-γ were found only in animals that received MVs carrying PUUV Gn/Gc proteins, as compared to MVs carrying Katushka2S. This data indicates that MVs treatment could stimulate cytokine production in the host.

## 4. Discussion

HFRS, caused by orthohantaviruses, is an emerging infectious disease that is endemic in multiple countries and poses a significant public health threat [76]. The incidence rate of HFRS in the Republic of Tatarstan is the highest in Russia, with over 1000 cases registered annually [10]. PUUV is the orthohantavirus strain commonly isolated from patient and rodent samples collected in the Republic of Tatarstan [4]. Most patients are young, previously healthy individuals without co-morbidities [77,78,79]. HFRS hospitalization causes a decline in individuals’ productivity for several weeks, producing negative socio-economic impacts on the patient’s family [80,81,82]. Although the fatality rate is low, a HFRS-related death can place substantial financial and personal burdens on families. There are limited options for HFRS management, as no approved vaccines are available to prevent PUUV infections and their treatment remains focused on symptom management [9].

MVs were proposed as vehicles for the delivery of vaccine antigens, since they were shown to induce innate and adaptive immune responses [59,60,83,84]. The advantages of using MVs, as compared to conventional vaccines, are that they are less toxic and are capable of crossing natural biological barriers [85,86,87,88]. MV-based vaccines can specifically target mucosal sites, which often cannot be reached by injected vaccines [59]. This non-invasive administration of MVs could be used for mass vaccination programs, especially in remote and challenging environments [59]. Another advantage of MVs is their activity [62]. MVs derived from bacterial outer membranes were shown to be effective carriers of immunogenic proteins and were capable of attenuating keratitis caused by *Pseudomonas aeruginosa* [89]. MVs obtained from outer membranes of Vibrio cholerae, and Neisseria meningitides could prevent infectious challenge and block transmission [60,84,90]. Recently, Hirayama and Nakao described a series of methods for isolating MVs from *Porphyromonas gingivalis* and analyzed the immunogenicity of these vaccines after intranasal administration [91]. Interestingly, MVs derived from bioengineered non-pathogenic bacteria were used as vehicles for viral antigens [58]. Rappazzo et al. demonstrated that engineering E-coli Nissle to display the influenza A matrix protein 2 (M2e) upon outer-membrane MVs resulted in significantly elevated anti-M2e IgG titers and therefore protected against influenza A strain PR8 [58]. In another study, Carvalho et al. demonstrated the efficacy of MVs derived from normal gut microflora in eliciting a mucosal immune response [59].

Our study aimed to investigate the immunogenic potential of MVs isolated from mMSCs expressing structural hantaviral proteins and their combination. For the first time, we have demonstrated the immunogenicity of MVs carrying PUUV proteins in vivo. Our data shows MVs’ capability to induce a humoral immune response. MVs carrying PUUV N and Gn/Gc proteins induced an IgG antibody response at 14 and 28 days after delivery. Interestingly, MVs carrying the combined PUUV N and Gn/Gc proteins had a higher capacity to induce an antibody response as compared to those delivering individual viral proteins. In addition to anti-orthohantavirus antibodies, we found that MVs carrying PUUV proteins could stimulate the T lymphocytes that recognize virus antigens. The activation of a cellular immune response is an important feature of effective vaccines [19], as virus-specific cytotoxic T cells could protect the host by eliminating infected cells [92,93,94,95].

For the first time, we demonstrated that MVs could induce a T cell immune response to N protein peptides previously identified as containing immunogenic epitopes [96,97]. One of these peptides, N25 (241–255 aa), was shown to activate PUUV-specific CD8+ memory T cells [96]. Furthermore, this peptide has a HLA-B8-restricted epitope, and it was one of the most consistently recognized among HFRS convalescent donors. Our data also identified a peptide containing 241–255 aa of N protein as immunogenic and showed that it produced a long term T cell immune response, as it was still reactive 28 days after MVs vaccination. Although we demonstrated a T cell reactivity to Gn/Gc peptides, it was detected only on day 14 after MV injection, suggesting its short-lasting nature. The detection of reactivity to the N25 (241–255 aa) peptide indicates that the immune response induced by MVs carrying PUUV N protein would be similar to that found in HFRS convalescent individuals. Therefore, we believe that orthohantavirus proteins carrying MV vaccines show a potential for developing a protective immune response against PUUV infection.

We have demonstrated the possibility of mMSC-derived MVs carrying PUUV N and Gn/Gc proteins serving as a delivery vehicle to induce a specific innate and adaptive immune response. We have shown that MVs could deliver PUUV N and Gn/Gc proteins to cells in vitro, confirming their ability to transfer viral proteins. In addition to PUUV proteins, MVs carry cytokines and chemokines (IL-6, CCL2, CCL3, CCL5, CCL11, CXCL1, and GM-CSF) as they were also found in the cargo. These cytokines, when released at the delivery site, could attract leukocytes, stimulate phagocytosis, and PUUV protein processing by antigen presenting cells [98,99,100,101,102]. Some of these cytokines were shown to be activated in HFRS caused by PUUV, suggesting their role in the induction of immune response to this orthohantavirus [3,103]. These data demonstrate that carrying cytokines and chemokines together with PUUV proteins could contribute to the development of an immune response by MVs.

We have also found that MVs were activating cytokines in animals. We demonstrated an increased TNF-α serum level after treatment, with MVs carrying PUUV N protein at 14 days after treatment. TNF-α is essential for triggering an immune response, as it stimulates phagocytosis by macrophages, which are essential antigen-presenting cells [104]. In contrast, increased CCL11 and decreased CCL3 levels were found in mice treated with MVs carrying PUUV Gn/Gc proteins. These chemokines are chemoattractants that regulate leukocyte migration [105,106]. Interestingly, CCL11 was shown to bind to CCR3 receptors, which are expressed on eosinophils [107]. The expression of CCR3 was also demonstrated on T helper type 2 (Th2) lymphocytes [108]. This observation led to the suggestion that CCL11 plays a role in developing a Th2 immune response [109]. We found an increased level of anti-PPUV antibodies in mice treated with MVs carrying PUUV Gn/Gc. Although the stimulation of the humoral immune response requires the cooperation of multiple cytokines, we suggest that CCL11 could contribute to antibody production after treatment with PUUV Gn/Gc MVs. The decreased serum level of CCL3 could indicate an impairment of leukocyte migration to the site of the MVs’ injection. These leukocytes include monocytes, macrophages, and natural killer cells [110]. CCL3 also contributes to type 1 macrophages’ polarization [111]. Therefore, we suggest that low levels of CCL3 in mice treated with PUUV proteins containing MVs could contribute to the polarization of macrophages to type 2.

Twenty-eight days after treatment with PUUV containing MVs, increased serum levels of IL-6, IFNγ, G-CSF, and GM-CSF was found. These cytokines are growth factors, stimulating leukocyte proliferation and differentiation [98,112,113,114] and serving as chemoattractants [99,115]. IL-6 is a pleotropic cytokine with profound effects on CD4 T cell proliferation [116] and it could protect lymphocytes from activation-induced cell death [117]. IL-6 can also stimulate the differentiation of lymphocytes, shifting the balance towards the Th2 population [118,119]. In contrast, IFNγ is a cytokine stimulating Th1 lymphocytes [120]. Together, increased serum levels of IL-6 and IFNγ could indicate the stimulation of a Th2 (humoral) and Th1 (cellular) immune response after treatment with PUUV containing MVs. This assumption is supported by our finding of anti-PUUV antibodies and anti-PUUV IFNγ-producing T cells. Increased serum levels of G-CSF and GM-CSF could indicate the stimulation of granulocyte and monocyte proliferation [121].The secretion of G-CSF and GM-CSF could be stimulated by IL-6 [121,122]. Both G-CSF and GM-CSF stimulate phagocytosis [123], which could play a role in the uptake of MVs and the processing of their cargo. Therefore, we suggest that the increased production of cytokines and chemokines triggered by MVs’ cargo (PUUV proteins, cytokines, and chemokines) could facilitate the antigen recognition and immune response development.

In summary, our data provides evidence that MVs could be used for the delivery of PUUV proteins and can induce a humoral immune response to orthohantavirus proteins. PUU-specific T cell were generated after treatment with MVs carrying PUUV proteins. These data present, for the first time, MVs as vehicles for the delivery PUUV antigens for vaccination.

## Figures and Tables

**Figure 1 pharmaceutics-14-00093-f001:**
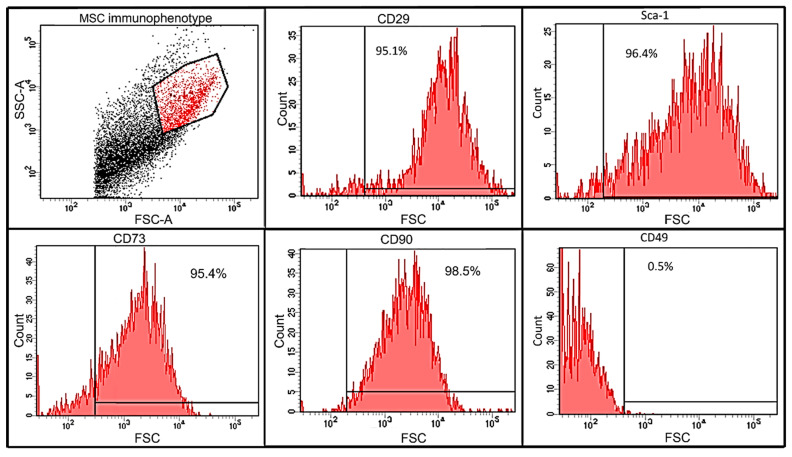
Immunophenotyping analysis of adipose-tissue-derived mMSCs by flow cytometry. Adipose-tissue-derived mMSCs were incubated in anti-mouse-CD29-PE, anti-mouse-Sca-1-AmCyan-A, anti-mouse CD90-BV421, anti-mouse-CD49-PE, and anti-mouse CD73-Alexa Fluor 647 antibodies. Cells were analyzed using flow cytometry on a FACS Aria III (Becton, Dickinson and Company, Becton Drive Franklin Lakes, Franklin Lakes, NJ, USA). A minimum of 300,000 events were collected for each sample. Results represent the percentage of cells expressing the surface markers.

**Figure 2 pharmaceutics-14-00093-f002:**
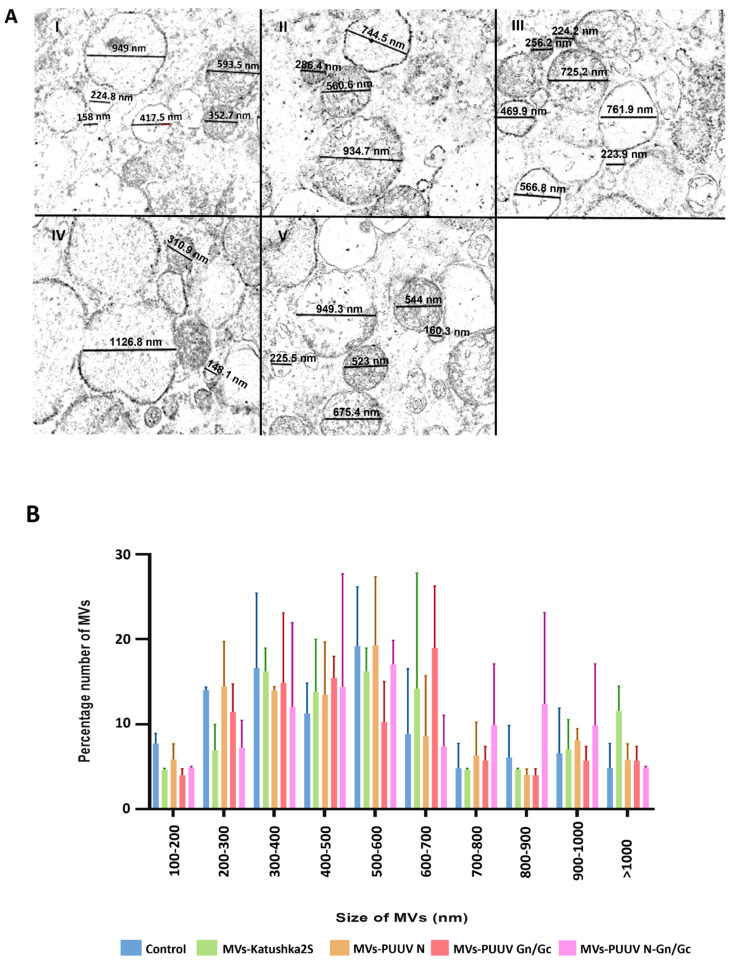
The structure and size distribution of MVs. (**A**)—TEM analysis was used to analyze the structure of mMSC-derived MVs (scale bar 1 µm). The diameter of the MVs (black lines) in each experimental group was calculated individually (five images per group) using ZEN 2 Blue Edition software. One example figure was demonstrated for each group: I—control MVs; II—MVs-Katushka2S; III—MVs-PUUV N; IV—MVs PUUV Gn/Gc; and V—MVs-PUUV N and Gn/Gc. (**B**)—The size distribution of MVs: control (blue); Katushka2S (green); PUUV N (orange); Gn/Gc (red); and a combination of N and Gn/Gc proteins (pink). MVs from non-transduced cells were used as the control. Data are presented as the percentage of MVs in each size range ± SD.

**Figure 3 pharmaceutics-14-00093-f003:**
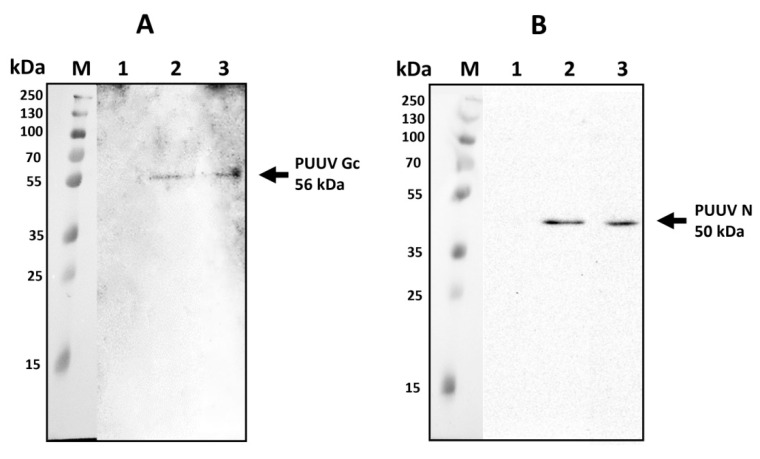
Western blot analysis of N and Gn/Gc protein load in MV cargo. Total proteins (10 µg) from MVs carrying PUUV N, Gn/Gc as well as a combination of PUUV N and Gn/Gc proteins were analyzed by Western blot. MVs from non-transduced mMSCs were used as the control. Proteins were probed with primary rabbit anti-N protein or mouse anti-Gc protein antibodies detecting PUUV N or Gc proteins, respectively. Antibody–antibody complexes were visualized using Clarity ECL substrate solution. (**A**)—PUUV Gc (56 kDa) in MVs. Lane A1—control; lane A2—MVs PUUV Gn/Gc; lane A3—PUUV N and Gn/Gc. (**B**)—PUUV N protein (50 kDa) in MVs. Lane B1—control; lane B2—MVs PUUV N; lane B3—PUUV N and Gn/Gc.

**Figure 4 pharmaceutics-14-00093-f004:**
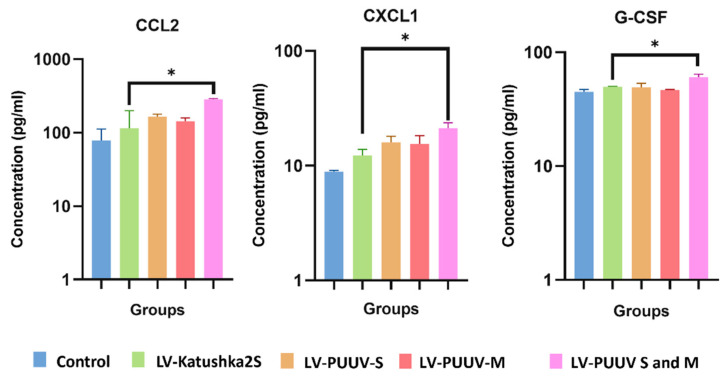
Cytokine and chemokine levels in mMSCs supernatant. mMSCs were transduced with LV-PUUV-S, LV-PUUV-M, as well as combined LV-PUUV-S and PUUV-M lentiviruses. Supernatants from non-transduced as well as cells transduced with LV-Katushka2S were used as the control. Data are represented as the median ± SD. * *p* < 0.05. A *p* value < 0.05 was considered statistically significant.

**Figure 5 pharmaceutics-14-00093-f005:**
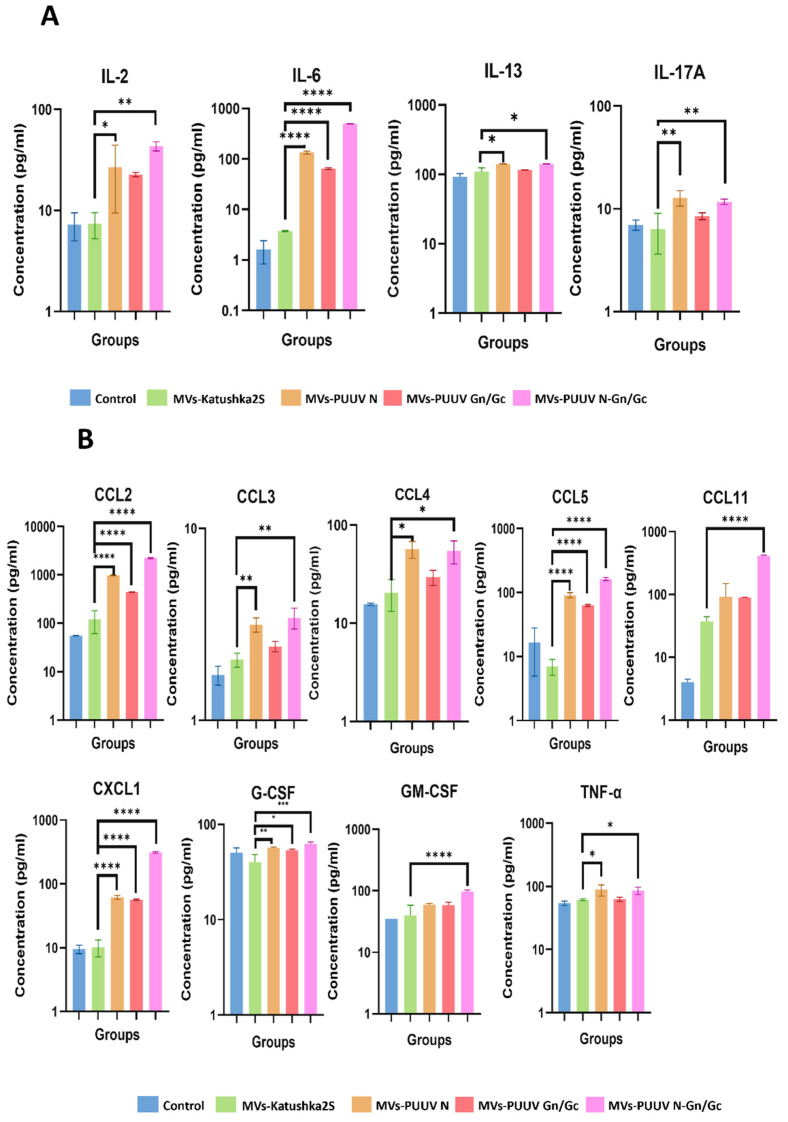
Cytokines and chemokine analysis of MVs cargo. Cytokine and chemokine levels were analyzed in MVs carrying PUUV N, PUUV Gn/Gc and combined PUUV N and Gn/Gc proteins, using Multiplex analysis. MVs generated from non-transduced as well as transduced with LV-Katushka2S mMSCs were used as control. MVs (50 µL in each well) with total protein (10 µg) were loaded into the well. (**A**)—Interleukin levels; (**B**)—cytokine and chemokine levels. Data is represented as median ± SD. * *p* < 0.05, ** *p* < 0.01, *** *p* < 0.005, **** *p* < 0.0001. A *p*-value < 0.05 was considered statistically significant.

**Figure 6 pharmaceutics-14-00093-f006:**
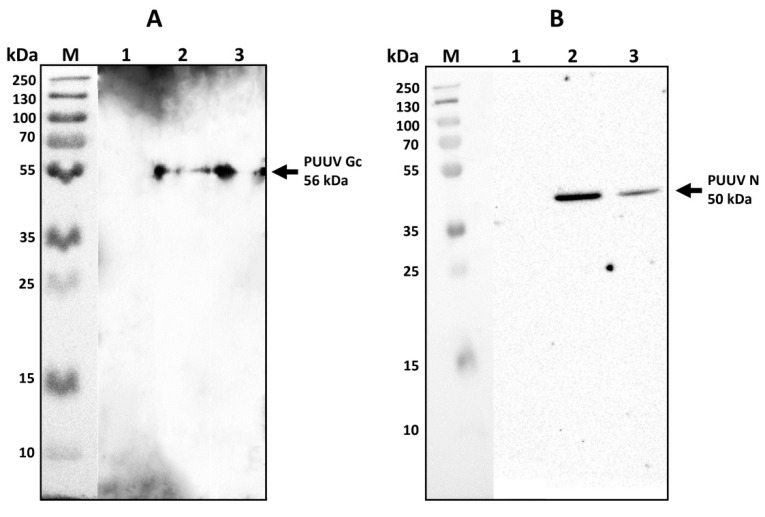
PUUV N and Gn/Gc proteins in A549 cells after MVs treatment. Proteins (10 µg) were loaded in each well and separated using electrophoresis. Primary rabbit anti-N protein or mouse anti-Gc protein antibodies were used to detect PUUV N or Gn/Gc proteins, respectively. PUUV proteins were visualized using Clarity ECL Substrate solution. (**A**)—PUUV Gc (56 kDa) protein in A549 cells treated with MVs carrying PUUV Gn/Gc (lane A2) or PUUV N and Gn/Gc (lane A3); (**B**)—PUUV N (50 kDa) protein in A549 cells treated with MVs carrying PUUV N (lane B2) or PUUV N and Gn/Gc (lane B3). A549 cells treated with MVs from non-transduced MSCs, were used as control (lane A1-B1).

**Figure 7 pharmaceutics-14-00093-f007:**
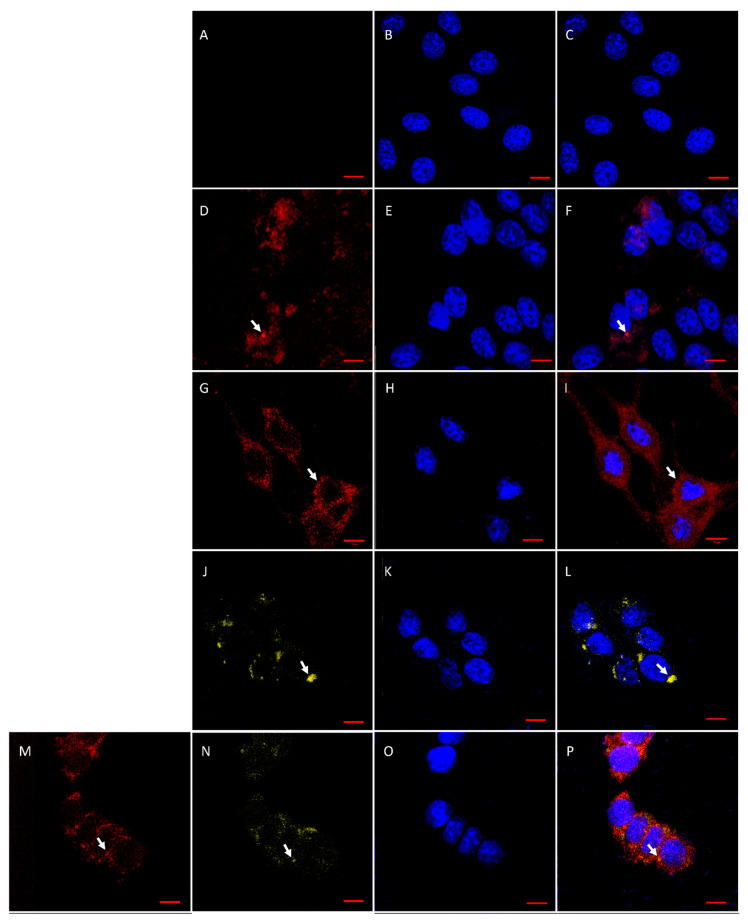
PUUV N and Gn/Gc protein detection in A549 cells treated with MVs. A549 cells (5 × 10^4^ cells/per well) were treated with MVs, containing PUUV N, Gn/Gc or their combination together with red fluorescent protein Katushka2S. Treated cells were incubated with primary rabbit anti-N protein or mouse anti-Gc protein antibodies followed by secondary donkey anti-rabbit IgG (H + L) Alexa Fluor 647 or donkey anti-mouse IgG (H + L) Alexa Fluor 555 to previous primary antibodies, respectively. (**A**)—A549 cells treated with MVs derived from non-transduced MSCs; (**B**)—The nuclei of A549 cells treated with MVs derived from non-transduced MSCs; (**C**)—Merged images (**A**,**B**); (**D**)—A549 cells treated with MVs-Katushka2S; (**E**)—The nuclei of A549 cells treated with MVs-Katushka2S; (**F**)—Merged images (**D**,**E**); (**G**)—A549 cells treated with MVs expressing PUUV N protein; (**H**)—The nuclei of A549 cells treated with MVs expressing PUUV N protein; (**I**)—Merged images (**G**,**H**); (**J**)—A549 cells treated with MVs expressing PUUV Gn/Gc protein; (**K**)—The nuclei of A549 cells treated with MVs expressing PUUV Gn/Gc protein; (**L**)—Merged images (**J**,**K**); (**M**)—A549 cells treated with MVs expressing PUUV N and Gn/Gc proteins; (**N**)—A549 cells treated with MVs expressing PUUV N and Gn/Gc proteins; (**O**)—The nuclei of A549 cells treated with MVs expressing PUUV N and Gn/Gc proteins; (**P**)—Merged images (**M**–**O**). Red fluorescence–PUUV N protein expression revealed by anti-rabbit AlexaFluor 647 antibodies. Yellow fluorescence–PUUV Gc protein expression revealed by anti-mouse AlexaFluor 546 antibodies. DAPI staining was used to demonstrate the nucleus. A549 cells treated with MVs derived from non-transduced MSCs, served as negative control. Scale bar 10 µm.

**Figure 8 pharmaceutics-14-00093-f008:**
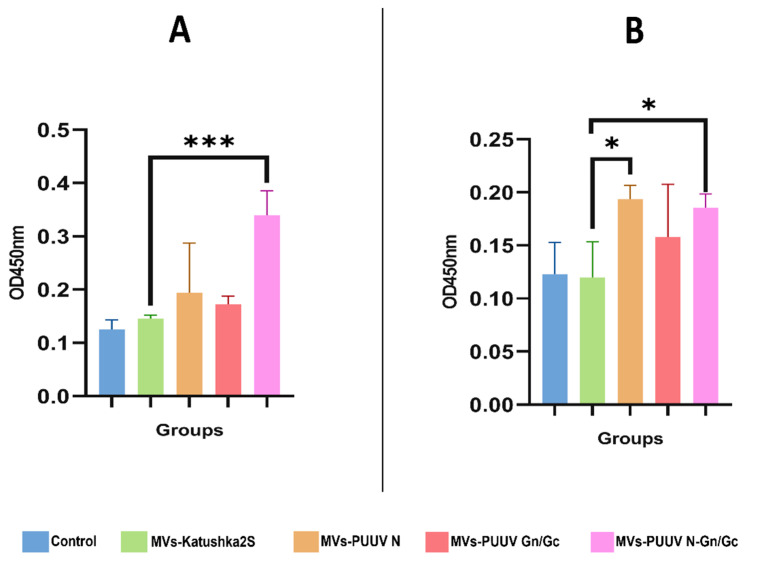
Serum anti-orthohantavirus IgG in mice treated with MVs carrying PUUV N, Gn/Gc as well as their combination. Serum samples were collected 14 and 28 after injection of MVs expressing PUUV N, Gn/Gc and combination of these proteins. Serum samples from the mice treated with 0.9% NaCl solution as well as from mice treated with MVs-Katushka2S were used as control. Anti-orthohantavirus proteins antibodies were detected using ELISA. (**A**)—anti-orthohantavirus antibody level at 14 days after MVs treatment; (**B**)—anti-orthohantavirus antibody level at 28 days after MVs treatment. Results are presented as OD_450_ values. Data is represented as median ± SD. * *p* < 0.05, ** *p* < 0.01, *** *p* < 0.005. *p* value < 0.05 was considered statistically significant.

**Figure 9 pharmaceutics-14-00093-f009:**
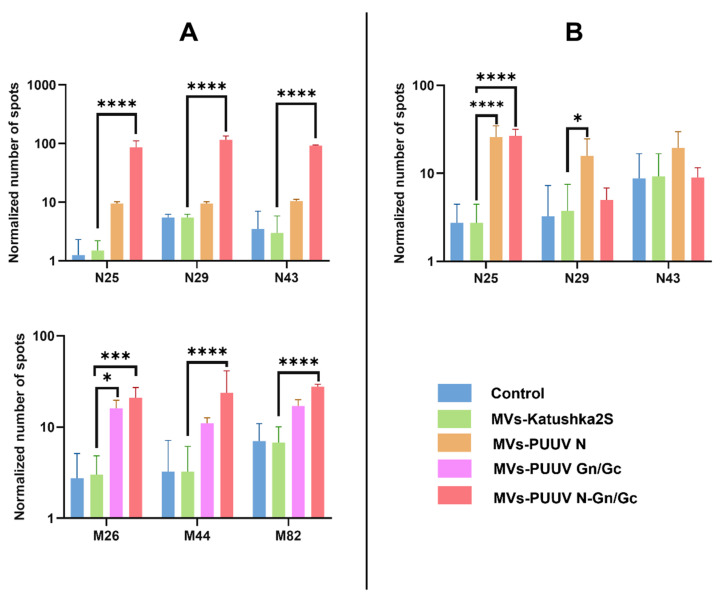
Analysis cytotoxic T lymphocytes at 14 and 28 days after MVs injection. ELISpot method was used for detection of INF-γ secretion by the activated T cells. Mice lymphocytes were isolated after treatment with MVs containing PUUV N, Gn/Gc, or their combination. Control lymphocytes were obtained from mice treated with 0.9% NaCl solution as well as from mice treated with MVs-Katushka2S. Lymphocytes (1 × 10^5^ cells) were placed into each well and treated with 1 μg of PUUV N25, N29, N43, M26, M44 or M82 peptides. Number of spots was counted to demonstrate the cytotoxic T cells activation. (**A**)—Cytotoxic T cell activation 14 days after MVs treatment; (**B**)—Cytotoxic T cell activation at 28 days after MVs treatment. Data is represented as median ± SD. * *p* < 0.05, ** *p* < 0.01, *** *p* < 0.005, **** *p* < 0.0001. A *p*-value < 0.05 was considered statistically significant.

**Figure 10 pharmaceutics-14-00093-f010:**
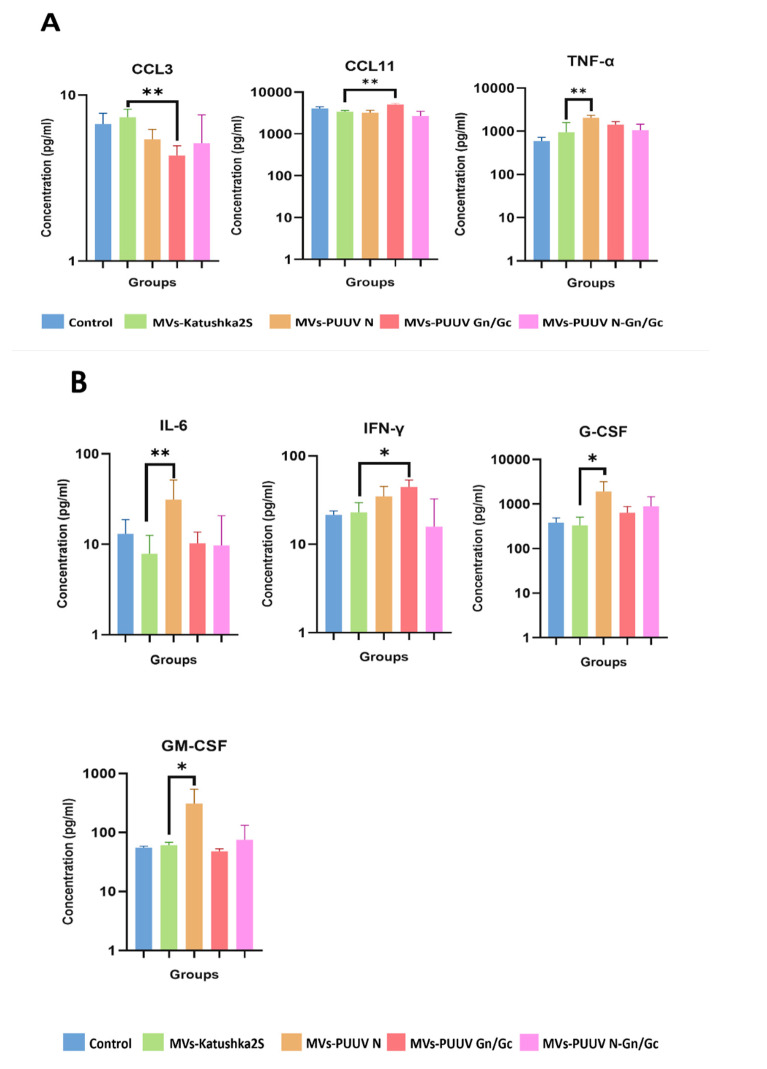
Serum cytokine analysis in mice treated with MVs at 14 and 28 days. Serum level of cytokines and chemokines was determined by using Multiplex method (BioRad). Serum was collected from mice treated with MVs containing PUUV N and Gn/Gc proteins, as well as their combination. Serum from mice treated with 0.9% NaCl solution as well as from mice treated with MVs-Katushka2S served as control. (**A**)—Serum cytokines and chemokines levels 14 days after MVs treatment; (**B**)—Serum cytokines and chemokines levels 28 days after MVs treatment. Data is represented as median ± SD. * *p* < 0.05, ** *p* < 0.01. *p* value < 0.05 was considered statistically significant.

**Table 1 pharmaceutics-14-00093-t001:** Peptides used in this study.

Peptides	Sequence (aa)	Position (aa)
PUUV N-25	EKECPFIKPEVKPGT	241–255
PUUV N-29	HVADIDKLIDYAASG	281–295
PUUV N-43	EIKVKEISNQEPLKI	424–438
PUUV M-26	FQGYYICLVGSSSEP	51–65
PUUV M-44	KFVCQRVDMDITVYC	431–445
PUUV M-82	YTRKACIQLGTEQTC	811–825

aa—amino acid.

**Table 2 pharmaceutics-14-00093-t002:** Lentiviruses used in this study.

Lentiviruses	Protein Expressed
LV-PUUV-S	PUUV N
LV-PUUV-M	PUUV Gn/Gc
LV-Katushka2S	Red fluorescent protein (Katushka2S)

## Data Availability

All data generated or analyzed during this study are included in this published article. The data that support the findings of this study are available from the corresponding author upon request.

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
