# Peer review of "Immunogenic Properties of MVs Containing Structural Hantaviral Proteins: An Original Study"

_pharmaceutics, 2022, doi:10.3390/pharmaceutics14010093_

Round 1

Reviewer 1 Report

This manuscript addressed the immunogenic properties of MVs. Some comments are listed below: 

  1. The Immunophenotype of mMSCs is determined using FACS with different markers. Why the authors select the small fraction of SSC-A needs to be described in the method. How about the control without treatment?
  2. The variation of the size distribution of TEM images for MVs is too big. It needs to provide more measurement for the size distribution.  
  3. The authors made the size distribution using TEMs, which is labelled with different samples in the one TEM image. The authors need to provide more information on how they determine which shape is MVS-PUUV N, MVS-PUUV Gn/Gc, etc.  
  4. In cytokine analysis, the authors need to provide further discussion on how the immune response last on different days (14 and 28 Days) and what the prediction of the immune response?  
  5. English needs to be improved and the grammar and spelling need to be further checked. For example: “Lack of specific treatment and vaccine makes HFRS management a challenge.’ in the abstract.

Reviewer 2 Report

This study by Shkair is straightforward and this reviewer can understand most parts of this study. One major concern is the the amount of MVs (proteins) for in vitro cell study and vaccination in vivo. As the authors can also notice that the immune responses are not such significant which might be caused by the actual abundance of the antigen displayed on MVs. This is the case also for the verification of MVs specifically for MVs produced by cells. The authors could perform a series of concentration to see if there is certain dose-dependent enhancement of the immune response from cells or individuals as used in this study.

Minor points:

1) Figure 2 is hard to follow. It is also required to explain the meaning of the differences of MV distribution among samples.

2) WB results are not informative. It might be necessary to show full/half length of the images used for this figure. Another suggestion for data presentation is to show all data points in all bar graph since the significance are not such obvious for many factors.

3) Language and writing should be improved for better understanding.

Reviewer 3 Report

The manuscript by Shkair et al. report an interesting strategy to develop a vaccine for hantavirus based on MV. The approach is interesting and straightforward, but the presentation of data is in some cases not good (see immunoblotting in Figure 3 and 6) or confused (see Figure 4 and 5). In addition, controls are not always appropriate, and these limitations are not reported in the Discussion. In addition, there is no discussion about other studies on MVs-based vaccines to better understand the context of the study and key information on the pathology should be improved.

Major points:

In the Abstract, authors wrote that “It has the highest incidence rate among zoonotic diseases in Russia.”. Could authors quantify the incidence? In the Introduction, could authors specify the incidence of severe and/or fatal disease? It is difficult to understand the health problems without these data

Authors reported in the Introduction about the poor efficacy of previously developed vaccines. Could they explain the hypothesis underlying this evidence? Is immunity acquired by infection protective?

Line 219 The title “TEM analysis of MVs.” Is not illustrative of the content of the section. Please specify better.

Line 223 The sentence “MVs were obtained 48 hours after transduction by using the cytochalasin B digestion method [49]” is misleading. Cytochalasin B treatment (not digestion) induce a higher release of microvesicles, but it is not a separation method to isolate microvesicles. The separation method should be specified.

It is difficult to gain insight MVs size from TEM image at a such high density. Could authors replace the image with another one with a lesser density of MVs?

It is not possible to understand in Figure 2 how the size of MVs was measured

Figure 3, showing immunoblotting analysis, is inappropriate: control and MVs-PUUV Gn/Gc or N-Gn/Gc should stay on the same lane filter. Besides, the enlargement is so high that is not possible to understand how many lanes were loaded. The same problem is present in Figure 6.

23 cytokines were analysed, but only a few of them were shown on Figure 4. Could authors comment on the behaviour of the other cytokines? More important, in Figure 5, authors reported cytokines increased in MVs cargo, but there is a lot of confusion, as not all of them were increased in all samples. Besides, the significance was calculated with respect to the control and not with respect to the MVs-Katushka, which is more appropriate, as the production of a transgenic protein may lead per se to an alteration of cytokine profile. Grouping the cytokines according to their behaviour and not just scattering them would help the reader.

The analysis of serum cytokine has an incorrect control, as the control is represented by mice injected with physiological solution and not by mice injected with MV from control or Katushka infected cells? The only comparison that is correct is between groups infected with different types of MVs.

Control based on the direct comparison with the injection of PUUV protein without the context of MVs have been not carried out. In the Discussion, authors should comment on that

In the Discussion, there is no mention of other studies aimed at developing vaccines based on MVs. Could authors insert their study in the context of the state of the art on this topic?

Minor points

Line 55 “…Multiple advantages of MVs as the delivery system”, please eliminate “the”

Line 57 “One these application was as a vehicle…” should be replaced by “One of these applications was as a vehicle…”

Line 66 “Additionally, activation of cytokines in mice treated with MVs taking PUUV N and Gn/Gc proteins was demonstrated.” Please specify the cytokines that were activated

Round 2

Reviewer 1 Report

I am happy with the current revision. 

Author Response

Thank you very much. Your considerations are sincerely appreciated.

Reviewer 2 Report

The authors revised the manuscript according to this reviewer's comment. One remaining question is that is it proper to use TEM to investigate the size? Usually, SEM or DLS is considered as a better presentation of intact MVs for size distribution analysis. 

Reviewer 3 Report

The manuscript by Shkair et al. has consistently improved. A few points to fix:

  • In Figure 5A, there is a significant difference for IL13 between control and MVs-Katushka2S. However, the meaning of this finding is difficult to catch. In addition, MVs-PUUV N have a higher value and a lower standard deviation as compared to MVs-Katushka2S, but it is not indicated as significantly different. Please check the graph
  • In the Discussion, examples of MV-based vaccines against bacteria are provided, but there are no examples related to MV-based vaccines against viruses. Could authors ass examples or comment on that?
